# Quantitative Proteomic Analysis of Plasma after Remote Ischemic Conditioning in a Rhesus Monkey Ischemic Stroke Model

**DOI:** 10.3390/biom11081164

**Published:** 2021-08-06

**Authors:** Siying Song, Linlin Guo, Di Wu, Jingfei Shi, Yunxia Duan, Xiaoduo He, Yunhuan Liu, Yuchuan Ding, Xunming Ji, Ran Meng

**Affiliations:** 1Department of Neurology, Xuanwu Hospital, Capital Medical University, Beijing 100053, China; skylarsong_2016@yeah.net (S.S.); qyguolinlin@163.com (L.G.); 20211280001@fudan.edu.cn (Y.L.); 2Advanced Center of Stroke, Beijing Institute for Brain Disorders, Beijing 100053, China; seadi-wu@163.com (D.W.); shijingff@163.com (J.S.); freefish0105@126.com (Y.D.); luhemunssy@163.com (X.H.); yding@med.wayne.edu (Y.D.); 3Department of China-America Institute of Neuroscience, Xuanwu Hospital, Capital Medical University, Beijing 100053, China; 4Department of Neurology, Beijing Geriatric Hospital, Beijing 100095, China; 5HuaDong Hospital, Fudan University, Shanghai 200040, China; 6Department of Neurosurgery, Wayne State University School of Medicine, Detroit, MI 48201, USA

**Keywords:** remote ischemic conditioning (RIC), rhesus monkey model, quantitative proteomics analysis

## Abstract

Background: Animal and clinical studies have shown that remote ischemic conditioning (RIC) has protective effects for cerebral vascular diseases, with induced humoral factor changes in the peripheral blood. However, many findings are heterogeneous, perhaps due to differences in the RIC intervention schemes, enrolled populations, and sample times. This study aimed to examine the RIC-induced changes in the plasma proteome using rhesus monkey models of strokes. Methods: Two adult rhesus monkeys with autologous blood clot-induced middle cerebral artery (MCA) occlusion underwent RIC interventions twice a week for five consecutive weeks. Each RIC treatment included five cycles of five minutes of ischemia alternating with five minutes of reperfusion of the forearm. The blood samples were taken from the median cubital vein of the monkeys at baseline and immediately after each week’s RIC stimulus. The plasma samples were isolated for a proteomic analysis using mass spectrometry (MS). Results: Several proteins related to lipid metabolism (Apolipoprotein A-II and Apolipoprotein C-II), coagulation (Fibrinogen alpha chain and serpin), immunoinflammatory responses (complement C3 and C1), and endovascular hemostasis (basement membrane-specific heparan sulfate proteoglycan) were significantly modulated after the RIC intervention. Many of these induced changes, such as in the lipid metabolism regulation and anticoagulation responses, starting as early as two weeks following the RIC intervention. The complementary activation and protection of the endovascular cells occurred more than three weeks postintervention. Conclusions: Multiple protective effects were induced by RIC and involved lipid metabolism regulation (anti-atherogenesis), anticoagulation (antithrombosis), complement activation, and endovascular homeostasis (anti-inflammation). In conclusion, this study indicates that RIC results in significant modulations of the plasma proteome. It also provides ideas for future research and screening targets.

## 1. Introduction

Remote ischemic conditioning (RIC) is an extremely simple, noninvasive, and cost-effective intervention. Former reviews have illustrated the multiorgan protective effect of RIC across the cardiovascular [1], renal [2], and cerebral systems [3]. The neuronal, humoral, and immunological signaling pathways, which all interact with one another, appear to play requisite roles in the mechanisms that underlie the RIC-related beneficial effects [4]. Our team firstly demonstrated the positive prognostic effects of RIC in a rhesus monkey model of ischemic stroke and in patients with acute ischemic stroke (AIS) after thrombectomy, ref. [5] intracranial atherosclerotic stenosis (ICAS) [6,7], and cerebral small vessel disease (CSVD) [8].

Few previous studies have evaluated the proteomic changes in response to RIC. We only found one prior animal study: Hibert et al. observed seven major modulated proteins, including haptoglobin and transthyretin, after one RIC intervention (testing five-min vs. 10-min sample times) in healthy rat models [9]. We also identified three clinical studies, each with different RIC intervention schemes, enrolled populations, and sample times. Hepponstall et al. demonstrated that fibrin beta, fibrinogen gamma, apolipoprotein A, and the acute phase response pathway were significantly regulated both 15 min and 24 h after a one-time RIC intervention in five healthy adult male volunteers [10]. Nikkola et al. enrolled 13 patients who had ruptured intracranial aneurysms and analyzed their DNA methylation and transcriptome profiles before and after a one-time RIC intervention [11]. Thorne et al. collected six pairs of human kidney transplant patients and observed the transient upregulation of several acute phase response proteins (SAA1, SAA2, and CRP) in the plasma at both one and five days post-RIC treatment [12].

There was too much heterogeneity among the previous clinical studies to draw consistent conclusions. Based on our precious clinical findings of the neuroprotective effects of RIC on cerebral vascular disease, we aimed to explore the peripheral blood proteomic responses after both short-term and long-term RIC in adult stroke monkeys. This is also the first study to dynamically evaluate proteomic changes in response to RIC.

## 2. Materials and Methods

### 2.1. Animals

Our study protocol was approved by the Animal Care and Use Committee of Capital Medical University (Beijing, China). All procedures were conducted in accordance with the Guide for the Care and Use of Laboratory Animals (National Research Council, Beijing, China), the Ministry of the Environment Guidelines for the Care and Use of Animals in Research, and the Guidelines for Proper Conduct of Animal Experiments (2006). All of the animal experiments were performed in accordance with the Animal Research: Reporting In Vivo Experiments (ARRIVE) guidelines.

Two healthy adult rhesus monkeys (*Macaca mulatta*) were included in this study. Both of them were free of TB, Shigella, Salmonella, Helminths, Ectoparasites, Entamaebahistolytica, and B virus. To increase the homogeneity, both animals were male, aged between 8 and 10 years, and weighed between 9 and 11 kg [13]. The monkeys were housed individually but in the same room and were given food, fruit, and water ad libitum.

### 2.2. Anesthesia and Intraoperative Management

The experimental animals were fasted for 12 h prior to undergoing anesthesia. Details of the anesthesia and intraoperative management were laid out in our previous study [14]. In brief, the experimental animals were anesthetized during the stroke model establishment, administration, and imaging examination. Ketamine (10 mg/kg) was administered intramuscularly to induce anesthesia, and propofol (300 lg/kg/min) was administered intravenously to maintain the anesthesia. All the experimental animals also underwent endotracheal intubation and mechanical ventilation during the stroke model establishment, and the relevant ventilator parameters were adjusted by an experienced anesthesiologist based on the actual situation. The anesthesia was lifted at the end of each procedure, when it was no longer needed. The major physiological parameters were continuously monitored during the procedure.

### 2.3. Stroke Model Establishment

The stroke model, which used middle cerebral artery (MCA)-M1 segment occlusion with an autologous blood clot, was developed by neuro-interventionalists based on our previous report [14]. The method of clot generation was run according to a recently published paper [15]. Briefly, we extracted 3–5 mL of femoral venous blood into a polyethylene catheter 24 h before the procedure. After being formed, a clot (of approximately 10 cm in length) was intercepted, transferred, and held at 4 °C. During the procedure of digital subtraction angiography (DSA), a Prowler-10 microcatheter (Codman, Johnson, MA, USA) with a Traxcess 0.010-inch guiding wire was introduced into the guiding catheter and navigated into the end of the M1 segment of the MCA. Following catheter placement, the clot was transferred into the microcatheter and flushed into the superior division of MCA-M2 segment with 1 mL of normal saline. The stroke was confirmed using MRI imaging and functional evaluations immediately following the clot placement.

### 2.4. Imaging Confirmation of Stroke Model

The MCA occlusion and newly formed ischemic stroke lesions at the M2 segment were confirmed with DSA and MRI scanning immediately after the operation (Figure 1a). MRI scanning was performed on a Magnetom Trio MRI Scanner (3.0 T; Siemens AG, Siemens Medical Solutions, Erlangen, Germany) with the following scan sequences: (1) T2-weighted imaging used the fast-spin echo method, TR = 4000 msec, TE = 100 msec, bandwidth = 200 Hz/pixel, FOV = 180 mm, slice thickness = 2 mm, 4 averages; (2) DWI, single-shot EPI, TR = 6600 msec, TE = 100 msec, bandwidth = 1002 Hz/pixel, FOV = 220 mm, slice thickness = 2 mm, 4 averages; and (3) MRA, TR = 20 msec, TE = 3.6 msec, bandwidth = 186 Hz/pixel, FOV = 220 mm, slice thickness = 1 mm, 1 average).

### 2.5. Limb RIC Intervention

RIC was achieved by the blocking arterial and venous blood flow of the two upper limbs for five min with 200-mmHg cuff pressure, followed by five min of deflation to restore perfusion. This cycle was automatically repeated for five bouts (10 min × five bouts) by the RIC device (patent number ZL200820123637.X, Beijing, China). RIC was initiated at the chronic stage (post-1.5 month) in the ischemic stroke monkeys and was repeated two times per week for a subsequent 5 weeks.

### 2.6. Blood Sampling

Blood samples were collected from monkeys through the femoral vein on the nonoperative side at six time points, including the baseline (prior to RIC), one week post-RIC, two weeks post-RIC, three weeks post-RIC, four weeks post-RIC, and five weeks post-RIC, respectively. The experimental schema is shown in Figure 1a.

### 2.7. Safety of RIC

To evaluate the safety of RIPC during acute ischemic stroke, we monitored the vital signs (including blood pressure, heart rate, and respiratory frequency) at the same time points when blood samples were collected. The local skin integrity was assessed by monitoring for any signs of swelling, erythema, and ecchymosis.

### 2.8. Sample Treatment

We analyzed the plasma samples from the two monkeys at all six time points (looking at 12 plasma samples in total). One hundred microliters of plasma were precipitated by adding PEG 6000 to a final concentration of 12% in order to deplete the abundant plasma proteins. Pierce^TM^ Top2 Abundant Protein Depletion Spin Columns (Thermo Fisher Scientific, Inc., Waltham, MA, USA) were used to pretreat the plasma samples in order to decrease the abundant albumin and antibody components of the human plasma samples in preparation for the mass spectrometry (Figure 1b).

### 2.9. Liquid Chromatography Tandem Mass Spectrometry (LC-MS/MS)

The samples were denatured at 95 °C, and trypsin was added at an enzyme:protein ratio of 1:30 (50 μL of 0.3-μg trypsin). The samples were then incubated overnight at 37 °C. The following day, the filters were inverted, centrifuged, and, to ensure the maximum yield, the digested peptides were collected with acetonitrile washing steps (6%, 9%, 12%, 15%, 18%, 21%, 25%, 30%, and 35%) at pH 10 using a self-made C18 conversion cartridge (Beijing Genomics Institute, Beijing, China). Finally, the samples were merged into three parts.

The eluted samples (500 ng × 3) were dried using a vacuum concentrator (Speedvac, Eppendorf, Thermo Fisher Scientific, Inc., Waltham, MA, USA) and resuspended with 5 μL 0.1% trifluoroacetic acid (TFA), followed by mixing in a matrix consisting of a saturated solution of α-cyano-4-hydroxy-trans-cinnamic acid in 50% acetonitrile (ACN) and 0.1% TFA at a 1:1 ratio. The mixture (1 μL) was spotted onto a stainless-steel sample target plate.

Peptide MS and MS/MS were performed on an nLC-Easy1000-Orbitrap Fusion system mass spectrometer (Applied Biosystems; Thermo Fisher Scientific, Inc., Waltham, MA, USA). Buffer A was 0.1% formic acid (FA), and buffer B consisted of ACN in 0.1% FA. The samples were separated on an ReproSil-Pur Basic C18 column (120 mm × 150 μm, 1.9-μm particle size; Dr. Maisch GmbH) over a gradient of 5–31% ACN in 0.1% FA (buffer B) from the beginning to 69 min and then over a 75% buffer from 70 min to 75 min. The flow rate was 250 nL/min. The mass spectrometer was operated in the data-dependent analysis (DDA) mode for automated switching between MS (MS1) and MS/MS (MS2) acquisitions. Full MS survey scans were acquired from 300 to 1400 m/z at a resolution of 12,000, and the top 12 most abundant precursor ions were selected for high collision energy dissociation (HCD) fragmentation. The MS2 fragment ion detection resolution was also set to 12,000.

The proteins were successfully identified based on a 95% or higher confidence interval of their scores in the Mascot v2.3 search engine (Matrix Science, Ltd., London, UK) using the Mascot and NCBI protein databases.

### 2.10. Proteomics Data Analysis

The MS raw data were examined using MaxQuant software (v1.3.1). The search parameters were: trypsin with two missed cleavages was allowed, oxidation (M) and deamidation (N and Q) were set as the variable modifications, and carbamidomethylation (C) was set as the fixed modification. Only unique and razor peptides were quantified. The identified peptides were processed using Perseus software (v1.5.4.1), which filtered out the contaminants and false-positive identifications (decoys). The label-free quantitation (LFQ) values were extracted from the MS spectra in MaxQuant using the MaxLFQ algorithm [15]. The accurate protein abundance was calculated from the sum of all the peptide intensities (the maximum detector peak intensity over the protein elution profile). MaxLFQ normalized the data to account for the variability of the quantifiable peptides across all the samples. The data were also log_2_-transformed.

For the proteomics data analysis and generation of the volcano plots, we used parametric one-way ANOVA tests (assuming a normal distribution of the quantitative mass spectrometry data) to compare between the different time points (log_2_ of the difference in the total intensities per protein).

We used three databases for the functional annotation and classification of the assembles’ unique sequences. (1) The Gene Ontology (GO) database was used to facilitate the biological interpretation of the identified proteins in these studies. The differentially expressed proteins in GO were divided into three categories: the biological process (BP), molecular function (MF), and cellular component (CC). (2) A pathway analysis was performed using the Kyoto Encyclopedia of Genes and Genomes (KEGG) database. The KEGG pathway database can facilitate a systematic, network-based understanding of the molecular interactions between genes [16]. A pathway-based analysis enables the further understanding of those biological functions. (3) EuKaryotic Orthologous Groups (KOG) classification was used to further evaluate the effectiveness of the annotation process [17].

## 3. Results

### 3.1. Profiling Differential Plasma Protein Expression between Baseline and Post-RIC Groups

We drew blood samples from the two monkeys at six time points: baseline, one week post-RIC, two weeks post-RIC, three weeks post-RIC, four weeks post-RIC, and five weeks post-RIC. We used a sum of these 12 samples to analyze for the differential proteomic analysis. Figure 2 shows the high correlations of the protein abundance levels between the two monkey models at all the time points. This finding indicates that the further differential proteomic analyses are reliable.

The differential analysis revealed 50 significantly modulated proteins between the baseline and various post-RIC time points, with 28 upregulated and 22 downregulated proteins (Figure 3A and Appendix A). The differential proteins are visualized in a volcano plot (Figure 3B).

### 3.2. Functional Annotation and Classification of the Assembled Unique Sequences

#### 3.2.1. GO Analysis

GO assignments can provide standardized vocabulary for assigning the functions of uncharacterized sequences. Thus, we used this method to classify the functions of the predicted genes. We compared the proteins that showed differences between the baseline and post-RIC time points (Table 1 and Appendix A). With each of the three categories of GO classification, the dominant subcategories were “cellular process”, “biological regulation”, and “metabolic process” among the biological processes; “cellular anatomical entity” among cellular components; and “binding” among the molecular functions. That is, the major GO classifications involved in the annotated sequences controlled the basic biological regulatory and metabolic mechanisms. Additionally, changing tendencies were found among the differential proteins at different post-RIC times. The total amount of upregulated proteins decreased to the lowest level at week two post-RIC and then gradually increased in weeks three and four (Figure 4A). This could be a reaction to the RIC stimulus. However, at week five, the number of upregulated proteins slightly decreased. The same tendency could also be observed in the dominant subcategories. In terms of the downregulated proteins, the scatter plot with smooth lines had a “w” shape (Figure 4B). Lower levels of the downregulated proteins were found at weeks two and four, and higher levels were found at weeks one, three, and five.

#### 3.2.2. KOG Analysis

KOG classification was used to further evaluate the effectiveness of the annotation process and the completeness of the transcriptome library. Differential proteins were identified and classified based on the KOG functional categories (Appendix A). We observed seven proteins that were significantly modulated after RIPC. These proteins were involved in the defense mechanisms, posttranslational modifications, chromatin structure and dynamics, and lipid transport and metabolism (Table 2).

#### 3.2.3. KEGG Analysis

The KEGG pathway database can facilitate a systematic understanding of the molecular interactions among the proteins and enables the further understanding of those biological functions. The top 40 differentially enriched pathways between the baseline and post-RIC samples are presented in Appendix A. We selected six major enriched pathways involving lipid metabolism, long-term depression, and immune system regulation (Table 3).

## 4. Discussion

Our study is the first to use a rhesus monkey ischemic stroke model to evaluate significant RIC-induced modulations in a blood protein composition based on a quantitative proteomic analysis. A long-term RIC intervention for five weeks was conducted, and blood samples were taken every week to dynamically follow-up on the potential changes. We systematically performed a GO, KOG enrichment analysis, and KEGG pathway analysis. Several proteins and pathways were found to be significantly modulated in response to RIC in both the short term and long term.

The differential analysis revealed 50 significantly modulated proteins between the baseline and various post-RIC time points, with 28 upregulated and 22 downregulated proteins (Figure 3). As shown in Figure 4, we followed up the changing tendency of the significantly enriched proteins based on the GO analysis. In terms of the upregulated proteins, the amount of proteins related to the “cellular process”, “biological regulation”, and metabolic process decreased two weeks after the RIC intervention, then gradually climbed up during weeks three and four, and finally slightly decreased to a level that was lower than the baseline. For the downregulated proteins, the scatter plot with smooth lines of proteins related to the “cellular anatomical entity”, “cellular process”, and “binding” had a “w” shape. The lower levels were found at weeks two and four, and the peak was reached at week five. No prior study has demonstrated a dynamic proteomic analysis along these lines. Based on our results, we presumed a state of homeostasis, involving the cellular responses, biological regulation, metabolic processes, and cellular anatomical entity to binding, may be achieved five weeks after the RIC intervention. These findings may also guide the duration of RIC therapy in clinical settings.

Several proteins and pathways were significantly enriched five weeks after RIC. One of our major findings was that the proteins related to the lipid metabolism and cholesterol metabolism pathways were largely regulated. We identified that Apolipoprotein A-II (Apo A-II) and Apolipoprotein C-II (Apo C-II) were upregulated after long-term RIC (i.e., a four-week time horizon). Apo A-II, as a component of high-density lipoprotein (HDL), helps initiate the process of HDL in removing low-density lipoprotein (LDL) from the body and has proven cardioprotective, anti-atherogenesis, and antioxidative effects [18]. Apo C-II plays a vital role in triglycerides (TG) metabolism by acting as a cofactor of lipoprotein lipase (LPL), the main enzyme that hydrolyses plasma TG [19]. These findings were consistent with numerous proteomic studies of RIC [9,10]. Moreover, we observed a high abundance of lecithin cholesterol acyltransferase (LCAT) and platelet activating factor acetylhydrolase (PAF-AH) as early as two weeks after RIC. These proteins respectively function as a key enzyme of HDL maturation [20] and a protector of HDL from oxidation [21,22]. Therefore, we assumed that both the short-term and long-term effects of RIC intervention could involve the anti-atherogenesis process by regulating the lipid metabolism in rhesus monkeys after an ischemic stroke, which may prevent a stroke recurrence.

The proteins related to coagulation were also regulated four weeks after RIC. The fibrinogen alpha chain, as a pro-coagulation factor, was downregulated, and alpha-2-macroglobulin, as an anticoagulation factor, was upregulated. Additionally, serpin, a super family of proteins with protease inhibition activity, was enriched for two weeks post-RIC. One of the most well-known antithrombosis proteins in this family is plasminogen activator inhibitor-1 (PAI-1) [23]. However, the levels of the Factor IX precursor increased. Therefore, the results were inconclusive regarding if RIC could induce anticoagulation effects (as opposed to pro-coagulation effects). Future studies with more samples and longer follow-up periods are suggested to help resolve these lingering questions.

Emerging evidence suggests that the induction of endogenous protection via RIC can be partially attributed to the modulation of immunoinflammatory responses. Proinflammation cytokines, such as IL-6 [24], IL-10 [25], and TNF-α [26], have been shown to be elevated in the rat models of stroke that were subjected to a pretreatment of RIC. In this study, a complement response was enriched since week three. Complement C3, C4b-binding protein beta chain, and Complement C1 subcomponent were upregulated, indicating activation of the classical pathway of the complement system in response to RIC stimulus. C-type lectin, functioning in innate and adaptive antimicrobial immune responses, had a high abundance one week after RIC. These findings suggest that early changes in the complement response may also contribute to the mechanisms underlying RIC-mediated neuroprotection.

Another novel finding was that RIC could also induce the enrichment of basement membrane-specific heparan sulfate proteoglycan (HSPG) after week 3. HSPG is primarily known as a key component of the extracellular matrix of cartilage, which is essential for normal growth plate development and long bone growth [27]. Moreover, HSPG also contributes to vascular homeostasis by (1) forming a vascular extracellular matrix to maintain the endothelial barrier function, (2) inhibiting smooth muscle cell proliferation, and (3) promoting growth factor activity to stimulate endothelial growth and regeneration [28]. Thus, RIC may also induce endovascular protective effects.

## 5. Conclusions

Based on the quantitative proteomic analysis, we found that RIC induced multiple protective effects, involving pathways related to lipid metabolism regulation (anti-atherogenesis), anticoagulation (antithrombosis), complement activation, and endovascular homeostasis. The major induced biological processes (such as lipid metabolism regulation and anticoagulation responses) started as early as two weeks following the initial RIC intervention. Complement activation and protection of the endovascular cells came after week three. At the end of the five-week-long RIC intervention, we presumed that a balanced state was achieved after the dynamic change of the protein modulations. In conclusion, this study indicates that RIC results in significant modulations of the plasma proteome and provides inspiration for future research, as well as potential screening targets for future works.

## 6. Limitations

Although this study, for the first time, finished a five-week long RIC intervention on stroke monkeys and dynamically monitored the changes of the plasma proteomics on weekly basis, the sample size was still small. The proteins and pathways identified in this study need to be confirmed by other experimental models and in other species before these results can be further explored in clinical studies. Moreover, a proteins/pathway interrelationship analysis can be conducted if the sample size is larger.

## Figures and Tables

**Figure 1 biomolecules-11-01164-f001:**
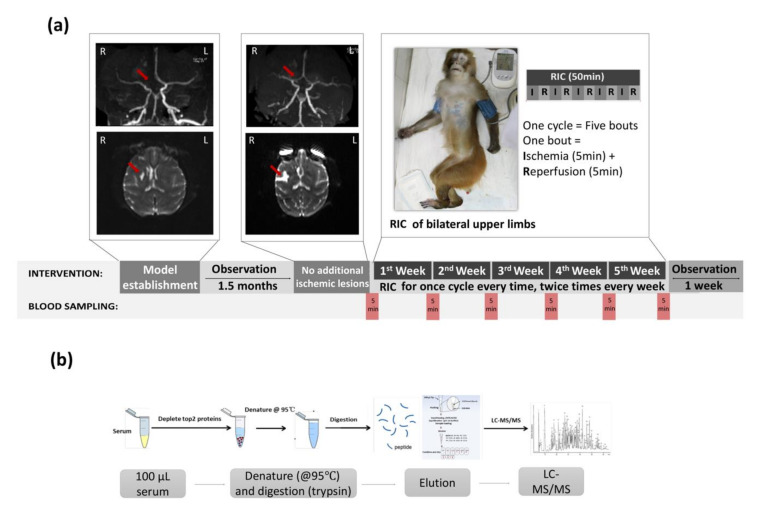
(**a**) Sample collection scheme and model establishment. The images of rhesus monkeys with acute right middle cerebral arterial occlusion-mediated stroke are presented at the time of model establishment (occlusion at the MCA-M2 segment (MRA, red arrow) and new temporal lobe infarctions 3 h after MCA-M2 occlusion (DWI, red arrow)) and at 1.5 months after MCA-M2 occlusion (partial recanalization of the occlusion (MRA, red arrow) and chronic temporal lobe infarction of the monkey (DWI, red arrow)). RIC, remote ischemic conditioning. (**b**) Proteomics analysis workflow.

**Figure 2 biomolecules-11-01164-f002:**
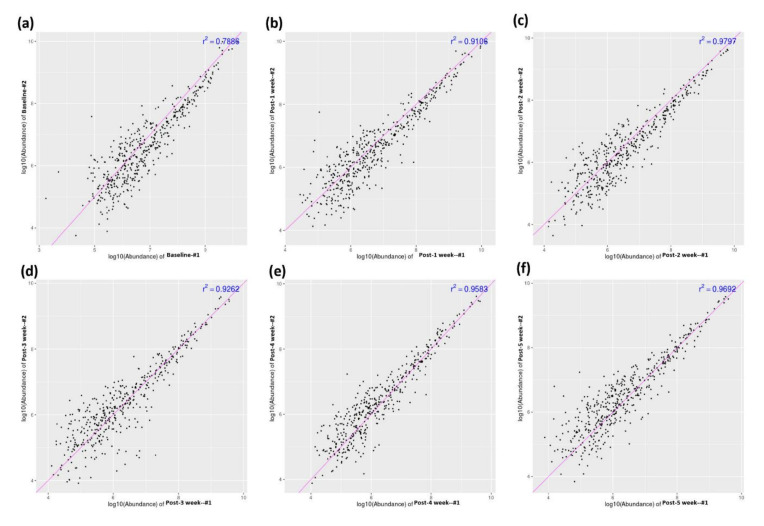
Correlation analysis of the protein abundance between the two monkey models. Baseline RIC (**a**), post-one week of RIC (**b**), post-two weeks of RIC (**c**), post-three weeks of RIC (**d**), post-four weeks of RIC (**e**), and post-five weeks of RIC (**f**).

**Figure 3 biomolecules-11-01164-f003:**
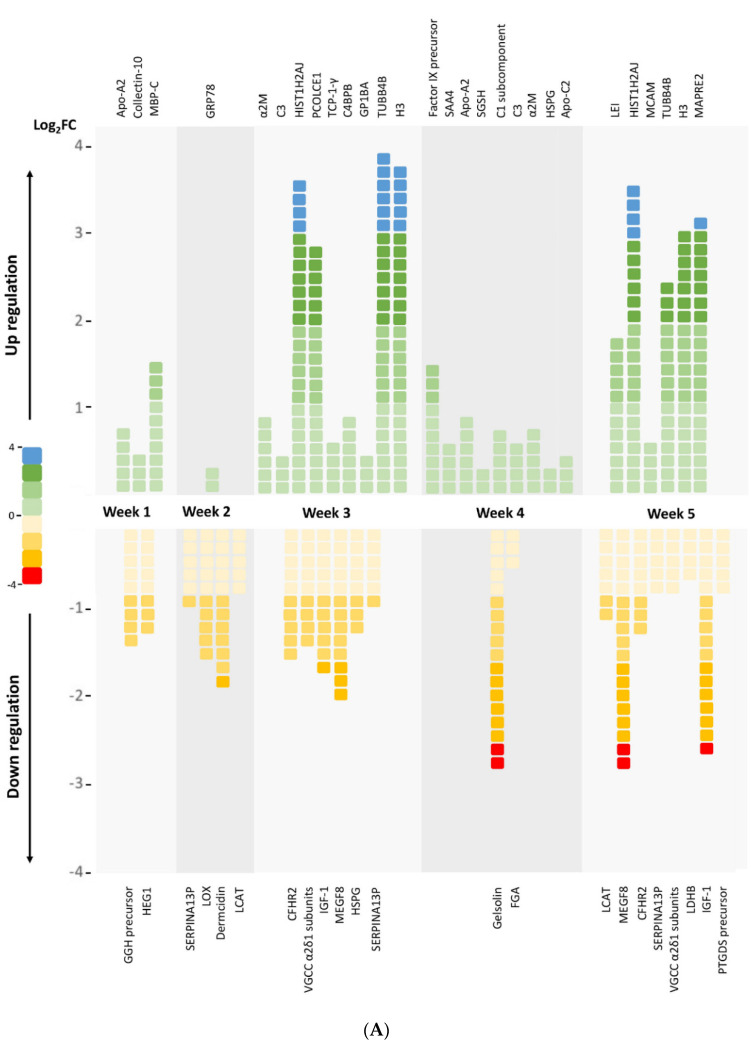
(**A**) Profiling of the differential plasma protein expressions between the baseline and the post-RIC groups. The six groups are: baseline (i.e., no intervention except blood sampling), post-one-week RIC, post-two-weeks RIC, post-three-weeks RIC, post-four-weeks RIC, and post-five-weeks RIC. The blood was sampled weekly after that week’s RIC therapy. FC = fold change; Apo-A2 = apolipoprotein A-II; MBP-C = Mannose-binding protein C; GRP78 = 78-kDa glucose-regulated protein precursor; α2M = alpha-2-macroglobulin; C3 = complement C3; HIST1H2AJ = histone cluster 1, H2aj; PCOLCE1 = procollagen C-endopeptidase enhancer 1; TCP-1-γ = T-complex protein 1 subunit gamma; C4BPB = C4b-binding protein beta chain; GP1BA = platelet glycoprotein Ib alpha chain; TUBB4B = tubulin beta-4B chain; H3 = histone H3; Factor IX precursor = coagulation factor IX precursor; SAA4 = serum amyloid A-4 protein; SGSH = N-sulphoglucosamine sulphohydrolase; C1 = complement C1s subcomponent; HSPG = basement membrane-specific heparan sulfate proteoglycan core protein; Apo-C2 = apolipoprotein C-II; LEI = leukocyte elastase inhibitor; MCAM = melanoma cell adhesion molecule cell surface glycoprotein MUC18; MAPRE2 = microtubule-associated protein RP/EB family member 2; GGH precursor = gamma-glutamyl hydrolase precursor; HEG1 = protein HEG homolog 1; SERPINA13P = putative serpin A13; LOX = protein-lysine 6-oxidase; LCAT = phosphatidylcholine-sterol acyltransferase; CFHR2 = complement factor H-related protein 2; VGCC α2δ1 subunits = voltage-dependent calcium channel subunit alpha-2/delta-1; IGF-1 = insulin-like growth factor I; MEGF8 = multiple epidermal growth factor-like domains protein 8; FGA = fibrinogen alpha chain; LDHB = L-lactate dehydrogenase B chain; PTGDS precursor = prostaglandin-H2 D-isomerase precursor. (**B**) Volcano plot demonstrating the differential plasma protein expressions between the baseline and post-RIC groups. Volcano plot showing the plasma protein level changes between: baseline RIC and post-one-week RIC (**a**), baseline RIC and post-two-week RIC (**b**), baseline RIC and post-three-week RIC (**c**), baseline RIC and post-four-week RIC (**d**), and baseline RIC and post-five-week RIC (**e**). *X*-axis: protein level difference indicated by the log2 fold change, *Y*-axis: statistical significance indicated by the log10 (*p*-value). In total, five, five, 15, 11, and 14 proteins had fold changes greater than 2 (at a *p*-value < 0.05) for (**a**–**e**), respectively. The upregulated proteins are highlighted in red, and the downregulated proteins are colored green.

**Figure 4 biomolecules-11-01164-f004:**
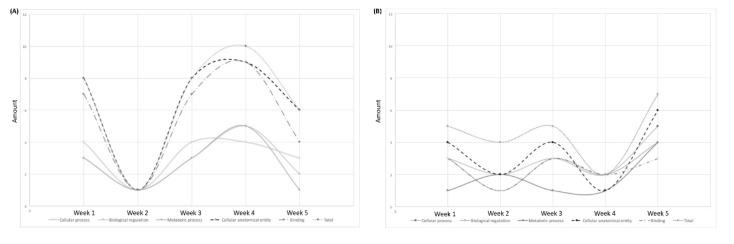
Amount of significantly regulated proteins during the five-week RIC intervention. (**A**) Upregulated proteins and (**B**) downregulated proteins.

**Table 1 biomolecules-11-01164-t001:** The major upregulated/downregulated proteins after RIC based on the GO analysis.

Protein Name	Modulation	Main Function	Major Changed Time Post-RIC
Apolipoprotein A-II	Upregulated	Lipid metabolism, antioxidative, and antiatherogenesis	Week 1; Week 4
Apolipoprotein C-II isoform X1	Upregulated	Lipid metabolism	Week 4
Complement C3	Upregulated	Humoral immunity	Week 3; Week 4
C4b-binding protein beta chain isoform X2	Upregulated	Humoral immunity	Week 3
Complement C1s subcomponent	Upregulated	Humoral immunity	Week 4
Histone cluster 1, H2aj	Upregulated	Chromatin structure and dynamics	Week 3; Week 5
Histone H3.1t-like	Upregulated	Chromatin structure and dynamics	Week 3; Week 5
Fibrinogen alpha chain	Downregulated	Hemostasis, platelet aggregation, and proinflammation	Week 4
Factor IX precursor	Upregulated	Hemostasis	Week 4
Alpha-2-macroglobulin	Upregulated	Fibrinolysis inhibition, anticoagulation, and carrier protein	Week 3; Week 4
Glucose-regulated protein precursor	Upregulated	Signaling, proliferation, invasion, apoptosis, inflammation, and immunity	Week 2
Insulin-like growth factor I isoform X3	Downregulated	Growth-promoting and DNA synthesis regulation	Week 3; Week 5
Putative serpin A13	Downregulated	Serine-type endopeptidase inhibitor activity	Week 2; Week 3

**Table 2 biomolecules-11-01164-t002:** Main functions of the enriched proteins after RIC based on the KOG analysis.

Protein Name	KOG ID	Functional Categories	Function Class Definition	Major Changed Time Post-RIC
Serpin	KOG2392	Cellular process and signaling	Defense mechanisms	Week 2; Week 3; Week 5
Basement membrane-specific heparan sulfate proteoglycan (HSPG) core protein	KOG3509	Cellular process and signaling	Posttranslational modification, protein turnover, chaperones	Week 3; Week 4
Histone 2A	KOG1756	Information storage and processing	Chromatin structure and dynamics	Week 3; Week 5
Histones H3 and H4	KOG1745	Information storage and processing	Chromatin structure and dynamics	Week 3; Week 5
Lecithin:cholesterol acyltransferase (LCAT)/Acyl-ceramide synthase	KOG2369	Metabolism	Lipid transport and metabolism	Week 2; Week 5
C-type lectin	KOG4297	Cellular process and signaling	Immunity and homeostasis	Week 1; Week 3
Attractin and platelet-activating factor acetylhydrolase	KOG1388	Metabolism	Lipid metabolism	Week 3; Week 5

**Table 3 biomolecules-11-01164-t003:** The major enriched pathways after RIC.

Pathway Name	Pathway ID	Major Changed Time Post-RIC
Cholesterol metabolism	mcc04979	Week 1; Week 2; Week 4
Glycerophospholipid metabolism	mcc00564	Week 2; Week 5
Long-term depression	mcc04730	Week 3; Week 5
MAPK signaling pathway	mcc04010	Week 3; Week 5
Complement and coagulation cascades	mcc04610	Week 3; Week 4
Phagosome	mcc04145	Week 3; Week 4

## Data Availability

The datasets generated during this study are available from the corresponding author upon reasonable request.

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
