# Peer review of "Quantitative Proteomic Analysis of Plasma after Remote Ischemic Conditioning in a Rhesus Monkey Ischemic Stroke Model"

_biomolecules, 2021, doi:10.3390/biom11081164_

Round 1

Reviewer 1 Report

Dear Authors,

The article entitled Quantitative proteomic analysis of plasma after remote ischemic conditioning in a rhesus monkey ischemic stroke model was very thoroughly prepared. All sections are correct and contain clear and precise wording. The research idea is exciting, and it has been duly implemented.

Generally, I don't have many comments.

1. The first and the most important concerns the lack of conclusions in the article's main text.
They are only in abbreviated form in the article abstract. The lack of precise conclusions makes it difficult to fully appreciate the quality of the results obtained in the study.

2. Besides, the authors wrote in the article: between Apo A-II, as a component of high density lipoprotein (HDL), 
helps initiate the process of HDL to removing bad types of cholesterol from the body, and have proven cardioprotective, anti atherogenesis, and antioxidative effects. - the word bad cholesterol should be removed because according to today's knowledge, the division into good cholesterol and bad cholesterol, apart from slang term, is very imprecise.

3. The article would be more interesting if the authors presented the obtained results schematically and showed the interrelationships between the changes in proteins/pathways resulting from remote ischemic conditioning.

Best regards

Author Response

Dear Reviewer 1,

Thanks so much for your valuable comments.

We have improved our manuscript based on your suggestions. All changed parts were marked in yellow.

  1. We added up in the conclusion part to make it more precise.
  2. We asked a native medical professionals to check our academic language again to ensure the precise use of all the medical terms.
  3. This is a really good suggestion to add more results on  the interrelationships between the changes in proteins/pathways. However, after careful discussion with the expert on proteomic analysis in our team (Dr. Siying Song), we decided not to add this part of result although it was very interesting, because proteins/pathways interrelationship analysis based on the small sample size of our study may lead to overestimated or even misestimated interpretations of the results. We preferred to add this limitation in the discussion part. Future proteomic study based on larger sample size is more suitable for interrelationship analysis.

Kind regards,

Ran Meng

Reviewer 2 Report

The study is well conducted and the interpretations are sounded. The authors explained adequately the limits of the study in the limitations section. Despite this fact and despite the fact that no observation was obtained for humans the work worth publication in Biomolecules as this will foster further research projects in the field.

The reviewer propose to accept the manuscript in the present form.

Author Response

Thanks so much for your kind comments. It's our great honor to publish our study in Biomolecules.